# A nonlinear correlation measure with applications to gene expression data

**Yogesh M. Tripathi**[1,2], **Suneel Babu Chatla**[3,4], **Yuan-Chin I. Chang**[1], **Li-Shan Huang**[3]*, **Grace S. Shieh** [1,5,6,7]*

1 Institute of Statistical Science, Academia Sinica, Taipei, Taiwan, 2 Indian Institute of Technology Patna, Bihta, India, 3 Institute of Statistics, National Tsing Hua University, Hsinchu, Taiwan, 4 Department of Mathematical Sciences, University of Texas at El Paso, El Paso, Texas, United States of America, 5 Bioinformatics Program, Taiwan International Graduate Program, Academia Sinica, Taipei, Taiwan, 6 Genome and Systems Biology Degree Program, Academia Sinica, National Taiwan University, Taipei, Taiwan, 7 Data Science Degree Program, Academia Sinica, National Taiwan University, Taipei, Taiwan

* lhuang@stat.nthu.edu.tw (LSH); gshieh@stat.sinica.edu.tw (GSS)

**Data Availability Statement:** All relevant data are within the paper and its Supporting Information files.

**Funding:** The Ministry of Science and Technology, Republic of China, provided funding for this study

## Abstract

Nonlinear correlation exists in many types of biomedical data. Several types of pairwise gene expression in humans and other organisms show nonlinear correlation across time, e.g., genes involved in human T helper (Th17) cells differentiation, which motivated this study. The proposed procedure, called Kernelized correlation ($K_c$), first transforms nonlinear data on the plane via a function (kernel, usually nonlinear) to a high-dimensional (Hilbert) space. Next, we plug the transformed data into a classical correlation coefficient, e.g., Pearson's correlation coefficient ($r$), to yield a nonlinear correlation measure. The algorithm to compute $K_c$ is developed and the R code is provided online. In three simulated nonlinear cases, when noise in data is moderate, $K_c$ with the RBF kernel ($K_c$-RBF) outperforms Pearson's $r$ and the well-known distance correlation (dCor). However, when noise in data is low, Pearson's $r$ and dCor perform slightly better than (equivalently to) $K_c$-RBF in Case 1 and 3 (in Case 2); Kendall's tau performs worse than the aforementioned measures in all cases. In Application 1 to discover genes involved in the early Th17 cell differentiation, $K_c$ is shown to detect the nonlinear correlations of four genes with *IL17A* (a known marker gene), while dCor detects nonlinear correlations of two pairs, and DESeq fails in all these pairs. Next, $K_c$ outperforms Pearson's and dCor, in estimating the nonlinear correlation of negatively correlated gene pairs in yeast cell cycle regulation. In conclusion, $K_c$ is a simple and competent procedure to measure pairwise nonlinear correlations.

## Introduction

Introduced in the 1990s, a microarray slide can simultaneously detect the expression of thousands of genes from a sample (e.g., from a tissue) under investigation, and cDNA microarray technology has become widely used following the seminal paper [1]. There are two types of experiments, namely static and temporal, in studies using microarrays [2]. In the former experiments, microarrays capture only a single moment of gene expression, whereas, in a

in the form of grants awarded to author YIC (MOST 108-2118-M-001-004-MY3), to L.S.H. (MOST 107-2118-M-007-002-MY2 and 107-2811-M-007-014, a postdoctoral fellowship supported SBC), and to GSS (MOST 107-2118-M-001-009-MY2 and 109-2118-M-001-001-MY2). Additionally, Academia Sinica, Taiwan, Republic of China provided a grant to GSS (AS-VTA-110-17). Indirect support was provided through salaries/facilities use by the home institutions of YIC, LSH and GSS. The funders did not have any additional role in the study design, data collection and analysis, decision to publish, or preparation of the manuscript. There was no additional external funding received for this study. The specific roles of these authors are articulated in the 'author contribution' section.

**Competing interests:** The authors have declared that no competing interests exist.

temporal experiment, the arrays are collected over a time course which allows the dynamic behavior studied. Because the regulation of gene expression is a dynamic process, it is important to identify changes in gene expression, as well as identify correlated genes and correlated mRNA and protein over time [2–4]. Thus, temporal experiments are commonly carried out in biological sciences [2], and there is extensive statistical literature on time course data analysis [5].

Exploring nonlinearity in biomedical data is gaining popularity in biomedical research methodologies [6]. Several types of pairwise gene expression in humans and other organisms show a nonlinear correlation across time, e.g., expression of paired genes involved in the early human T helper (Th17) cell differentiation, show nonlinear correlation across time (Fig 4 of [7]; Fig 7 of [8]), a phenomenon which motivated this study. Th17 cells have been demonstrated to play an important role in auto-immune disease and inflammation in humans. Moreover, time-course expression of genes involved in the yeast and human cell cycle are also nonlinear (see Fig 1A for details; [9, 10]). However, classical correlation coefficients such as the Pearson's correlation coefficient (Pearson's $r$) and the Spearman's rank correlation measure only the linear relationship between two variables (e.g., genes in this study). Fig 7 in [8] led us to hypothesize other novel Th17-specific genes can be identified by having the top-ranked absolute nonlinear correlations (across time) with *IL17A*. Similarly, novel cell cycle genes may be inferred through the top-ranked nonlinear correlations with a known marker gene, e.g., *HST3* in Fig 1A. This motivated us to develop a measure for the nonlinear correlation between two variables.

Székely and colleagues proposed distance correlation (dCor) as a measure of dependence and test for independence between two random vectors [11]. Dcor can be easily implemented in arbitrary dimensions and is widely cited. However, dCor is based on distance covariance, thus it ranges between 0 and 1, i.e., it can not be negative. Chen and colleagues [12] introduced a nonparametric test to detect nonlinear correlations of time-course gene expression data (GED). The maximal local correlation metric was shown to detect the nonlinear association of five time-point GED between *rd* mice and age-matched wild-type controls, while other correlation methods such as Pearson correlation could not. Later on, an empirical copula-based statistic (CoS) was developed to assess the strength of and test for independence between two random variables [13].

Here, we propose a procedure to measure the nonlinear correlation between two variables across time, using time-course microarray and RNA-seq gene expression data. To the best of our knowledge, this nonlinear measure is the first one that can quantify a negative correlation (a complementary pattern) of two variables by a negative value, e.g., *HST3-RAD51* in Fig 1A. This procedure first transforms nonlinear data on the plane (say, $x$ in $R^2$) via a function (called a kernel, usually nonlinear) to a high-dimensional (Hilbert) space (say, $\tilde{x}$, [14, 15]). A chosen kernel implicitly defines a transformation that conducts a "nonlinear transformation" of the original observation $x$ to $\tilde{x}$. Next, we plug the transformed data $\tilde{x}$ into a classical correlation coefficient, e.g., Pearson's $r$. Because this is a nonlinear transformation, the correlation computed of $\tilde{x}$ will be the nonlinear correlation of the original observation $x$; more details can be found in the Materials and Methods section. This nonlinear correlation coefficient, called kernelized correlation (denoted by $K_c$), can capture the nonlinearity across time between two variables or nonlinear relationships of any two pairwise variables in general. As indicated in [13], nonlinear correlations are prevalent in many applications, but not many have been developed. If any approach is developed, there will be many applications in data exploration, variable selection, and others. Thus, $K_c$ can also be applied to data exploration and variable selection. For example, of the human genome, genes with the top-ranked absolute $K_c$ values with *IL17A*, a known marker gene in early Th17 cell differentiation, may be of interest to biologists.

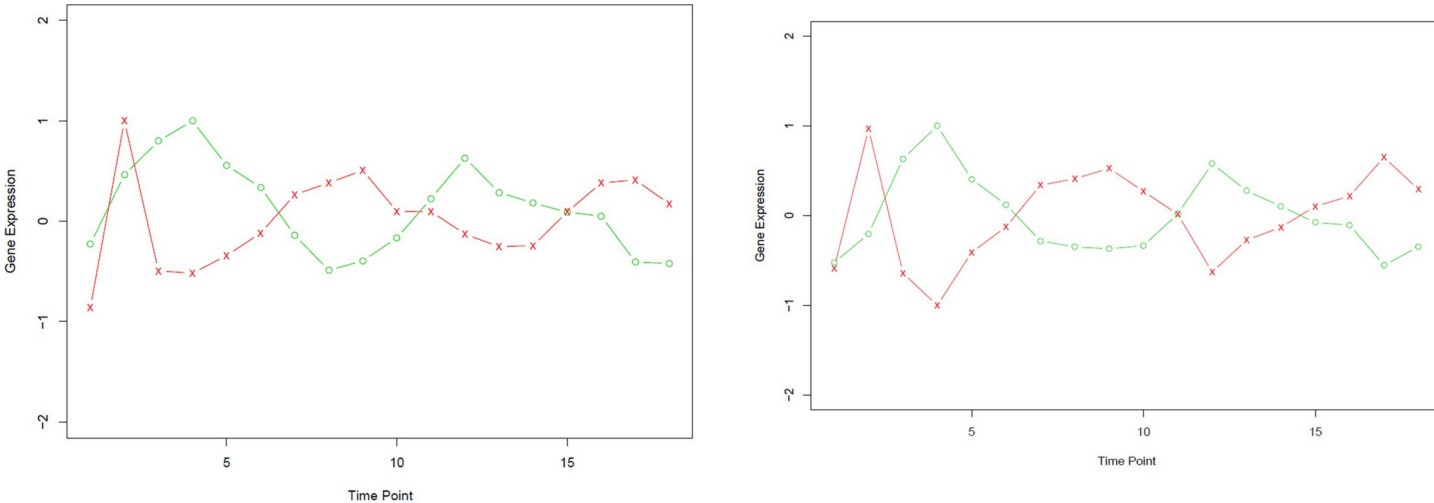

**Fig 1. A.** Original time-course expression of the gene pair *RAD51-HST3*, where the red cross (green circle) denotes expression levels of *RAD51* (*HST3*). **B.** Kernelized (using polynomial kernel of degree 2) expression of the gene pair *RAD51-HST3*.

$K_c$ is applicable to detect the correlation of any two genes over time within a single biological group, such as the pairwise correlations of ~800 cell cycle genes in yeast [9]. Furthermore, $K_c$ can also detect the correlation of one gene's expression over time between two groups. For example, pairwise correlation of each gene's expression in the mouse genome, profiled from age-matched wild type controls and *rd* mice with rod degeneration at five postnatal time points (6, 10, 14, 17, and 21 days of age) [11]. Although examples of short time-course experiments are demonstrated for $K_c$, as $K_c$ plugs transformed data into any correlation coefficient, it can be applied to long series of data in the same way as any classical correlation coefficient.

As an important application, the proposed $K_c$ has been demonstrated to uncover known and novel genes involved in the early differentiation of Th17 cells, by revealing genes having a nonlinear correlation with a known marker gene such as *IL17A*, whose expression is commonly used to assess the Th17 polarization efficiency ([8, 16, 17]). These uncovered genes are likely to be involved with the early Th17 cell differentiation. Thus, $K_c$ is a simple but efficient way to identify genes associated with early Th17 cell differentiation. Aijo and colleagues generated RNA-seq data to measure gene expression during early human T helper 17 (Th17) cell differentiation and T-cell activation (Th0) [8]. They let RNA-seq (count data) of a gene assume a negative binomial distribution with the mean following a Gaussian process at the *i*th time point of the *j*th replicate. Further, they employed an MCMC method to identify differential expression dynamics between Th17 and Th0; some selected identified differentially expressed genes were verified by qRT-PCR. The method was called DyNB, which extended maSigPro-GLM [18]. maSigPro-GLM is a package, that takes temporal dimension and correlation of RNA-seq time series into account, to detect differential expression of time-course RNA-seq data with or without replicates.

Furthermore, we applied $K_c$ to reveal the nonlinear correlation between pairs of cell cycle genes in yeast [9]. Proper regulation of the cell cycle is crucial to the growth and development of all organisms, and understanding this regulation is central to the study of many diseases. As shown in Fig 1A, the gene pair *RAD51-HST3* in yeast has a complementary pattern which suggests that their correlation is negative. The nonlinear correlation of *RAD51-HST3* is also supported by the PARE score -18.6, which is mainly based on the area enclosed by the two curves (Section 2.3, [19]). Although Pearson's *r* for *RAD51- HST3* is -0.50, it is not significant

($P$ = 0.203); see S2 Dataset for details). Thus, it does not reflect the negative nonlinear correlation between *RAD51* and *HST3*, as Pearson's *r* only measures linear association between the two genes (variables).

Although we demonstrate $K_c$ using time-course gene expression data, it can be applied to any two pairwise variables in general and other types of multi-dimensional data, e.g., proteomics and metabolomics data. In the Materials and Methods section, sources of data for applications are stated, the kernelized correlation is introduced and the algorithm provided. Furthermore, a code of $K_c$ to compute [n(n-1)]/2 pairwise correlations in a set of data consisting of n variables, where n could be 20,000 genes in a microarray, can be downloaded at http://staff.stat.sinica.edu.tw/gshieh/KC/Kc.html. In the Result section, $K_c$ is applied to measure nonlinear relationships between two experimental units (genes in the case of the motivating applications), using the time-course expression of paired genes. First, we compare $K_c$ to Pearson's correlation coefficient via a simulation study. Next, we show that $K_c$ reveals known and potential genes involved in the differentiation of Th17 cells in Application 1. As biological verifications are available in [8], we compare $K_c$ to dCor [11], in addition to DyNB [8] and a time point-wise analysis DESeq [20]. Furthermore, Pearson's correlation, dCor, and $K_c$ are applied to measure nonlinear relationships of gene pairs involved in the cell cycle of yeast [9] in Application 2, and their performances are compared. We close with some discussions and future directions.

## Materials and methods

### Data

In the Application section, we used publicly available data. Specifically, the time-course expression levels (RNA-seq; in normalized read counts) of eight genes involved in the differentiation of human Th17 cells, profiled in Th0 and Th17 cells were downloaded from [8]. Moreover, time-course microarray gene expression of six gene pairs was downloaded from [9]. Both datasets are available in the S1 and S2 Datasets, respectively.

### Transforming nonlinear data to a high-dimensional space via a kernel

Kernelization is known to help reveal nonlinear structures of data [15], provided that there is sufficient data. Here, we propose mapping the original nonlinear data in Euclidean space $R^2$ (e.g., the data in Fig 1) to a high-dimensional Hilbert space, then computing a linear correlation, e.g., Pearson's *r*, based on the kernelized data to result in the nonlinear correlation.

Suppose that $X$ and $Y$ are two random variables, and $x_1, \ldots, x_T$ ($y_1, \ldots, y_T$) are $T$ observations of $X(Y)$. Let $\vec{x} = (x_1, \ldots, x_T)'$ and $\vec{y} = (y_1, \ldots, y_T)'$ denote two $T \times 1$ vectors. For example, $T$ is the total number of time points in experiments, and $T = 5$ for immune genes [8]. Note that this kernelization is not limited to time-course data, it can be applied to any pairwise variables. In the following, we introduce the polynomial and Gaussian (or Radial Basis Function (RBF)) kernels, which are commonly used to transform the data into the high-dimensional Hilbert space.

**Definition.** A polynomial kernel of degree $d$ ($\geq 2$) is defined as $k(\vec{x}, \vec{y}) = (1 + <\vec{x}, \vec{y}>)^d$, where the inner product between $\vec{x}$ and $\vec{y}$ is defined as $<\vec{x}, \vec{y}> = \sum_{i=1}^{T} x_i \cdot y_i$.

**Definition.** A Gaussian (RBF) kernel is defined as $k(\vec{x}, \vec{y}) = \exp(-\gamma \| \vec{x} - \vec{y} \|^2)$ where the parameter $\gamma$ ($> 0$) is known as the inverse kernel width and the norm of $\vec{x}$ is defined as $\| \vec{x} \| = \sqrt{x_1^2 + x_2^2 + \cdots + x_T^2}$.

The following steps are executed to transform a given data set to a nonlinear high-dimensional space and compute the proposed kernelized correlation.

(i) Standardize the original gene expression data across time points of each variable (gene here).

(ii) Choose a kernel, e.g., Gaussian kernel.

(iii) Let $\vec{u}_i' = (x_i, y_i)$ and calculate the kernel matrix $\mathbf{K}$, where

$$\mathbf{K} = \begin{pmatrix} k(\vec{u}_1, \vec{u}_1) & k(\vec{u}_1, \vec{u}_2) & \dots & \dots & k(\vec{u}_1, \vec{u}_T) \\ k(\vec{u}_2, \vec{u}_1) & k(\vec{u}_2, \vec{u}_2) & \dots & \dots & k(\vec{u}_2, \vec{u}_T) \\ \dots & & \dots & \dots & \dots \\ \dots & & \dots & \dots & \dots \\ k(\vec{u}_T, \vec{u}_1) & k(\vec{u}_T, \vec{u}_2) & \dots & \dots & k(\vec{u}_T, \vec{u}_T) \end{pmatrix}$$

and $\mathbf{K}$ is a $T \times T$ symmetric and strictly positive definite matrix.

(iv) Center the above kernel matrix $\mathbf{K}$ by the following formula and denote the modified matrix by $\mathbf{K_c}$, where

$$\mathbf{K_c} = \left(\mathbf{I}_T - \frac{\mathbf{1}_T \mathbf{1}_T'}{T}\right) \mathbf{K} \left(\mathbf{I}_T - \frac{\mathbf{1}_T \mathbf{1}_T'}{T}\right),$$

$\mathbf{I}_T$ is a $T \times T$ dentity matrix and $\mathbf{1}_T$ is a vector of one's in $R^T$.

(v) Calculate $\mathbf{K_c}\vec{x}$ and $\mathbf{K_c}\vec{y}$ to result in the kernelized data corresponding to the original data. Next, we plug the associated elements of $\mathbf{K_c}\vec{x}$ and $\mathbf{K_c}\vec{y}$ into a correlation coefficient such as Pearson's $r$, to produce the proposed kernelized correlation coefficient.

Note that $K_c$ is nonlinear, thus the standardization of variables in Step 1 is essential, where standardization means the z-score of the original variable. Moreover, the values of kernelized data depend on the choice of the kernel and its associated parameter which is trained by cross-validation. Fig 1B is the kernelized gene expression of (*RAD51*, *HST3*) using the polynomial kernel of degree 2, which depicts the nonlinearity of the original data well. As an illustration, we use gene expression of the three-time points, $t = 2$, 3, and 4 of the gene pair *RAD51-HST3* in [9], to compute the corresponding $\mathbf{K}$ and $\mathbf{K_c}$ in the following. We also obtain $K_c = -1$ for *RAD51-HST3;* the expression of *RAD51* and *HST3* is depicted in Fig 1A.

$$\mathbf{K} = \begin{pmatrix} 1 & 0 & 0 \\ 0 & 1 & 0.988 \\ 0 & 0.988 & 1 \end{pmatrix} \quad \mathbf{K_c} = \begin{pmatrix} 0.886 & -0.443 & -0.443 \\ -0.443 & 0.227 & 0.216 \\ -0.443 & 0.216 & 0.227 \end{pmatrix}$$

## Statistical analyses

One sample t-test was used to test for the significance of all estimated correlation coefficients. All statistical tests were one-sided except where otherwise specified, and all analyses were conducted in the R software [21].

## Results

### Simulation study

Spellman and colleagues built a comprehensive catalog of yeast genes involved in the cell cycle [9], i.e., their mRNA levels vary periodically within the cell cycle. Specifically, the profiled mRNA levels of yeast genes using two-color cDNA microarray, using synchronized cell cultures, e.g., synchronized by $\alpha$ factor arrest. In the alpha set of microarray data, they profiled gene expression for two cell cycles at 18-time points, where time $t = 0,7,14,\ldots 119$ min. They identified the 800 cell cycle regulation genes by their correlations with genes known to be regulated by the cell cycle. As indicated in [9], these genes were regulated in a periodic way coincident with the cell cycle, for the adequate functioning of mechanisms that maintain order during cell division. In Fig 3(A) of [9], genes peaked at the G1 phase of the cell cycle and regulated similarly to the G1 cyclin *CLN2* were clustered. In particular, the similar-patterned *RNR1-SWE1* gene pair was studied by a time-lagged correlation and machine learning approach (Fig 1(a) of [19]).

We mimicked yeast cell-cycle genes in the alpha data set [9], e.g., *RNR1-SWE1*, to generate the expression of two genes across time. We generated pairwise gene expression in three cases representing different nonlinear relationships and compared the performances of six correlation measures. The six correlation measures studied are $K_c$ with the polynomial kernel of degree 2 and degree 3 (denoted by $K_c$-poly2 and $K_c$-poly3, respectively) and the RBF kernel ($K_c$-RBF), Pearson's correlation ($r$), Kendall's correlation ($\tau$) and distance correlation (dCor). Case 1 is composed of the time-course expression of two genes that follow the same sine function except with a $\pi/6$ time difference. In Case 2, the expression of two genes follows the same cosine function, but the magnitudes of gene 1 ($G_1$) are twice of gene 2 ($G_2$)'s. Identical time-course expression of two genes is generated in Case 3, except that $G_2$ has a location shift of 3 in the y-axis above $G_1$.

Specifically, time-course expression of paired genes with 18-time points was generated with 100 replications in each of the three cases. We mimicked yeast cell cycle genes in the alpha data set [9] to generate gene expression 7 minutes apart, i.e., $T_i = 0, 7, \ldots, 119$, using Eqs (1)–(6) in the following section. The error terms $\varepsilon_{1i}$ and $\varepsilon_{2i}$ were independent and generated from i.i.d. N (0,1), $i = 1, \ldots, 18$, and $c = 0.5, 1$, and 2 which provided small, medium to large random errors to mimic noises in real data. When $a = 0$ in Eqs (1)–(6), $G_1$ and $G_2$ are independent, by which we estimated the false positive rate (FPR) of the five correlation measures using 100 repeat simulations. The FPR is interpreted as the Type I error. When $a = 1$ in Eqs (1)–(6), we simulated 100 replicates to estimate the true positive rates (TPRs; statistical power) and summarize the mean (the estimate for nonlinear correlation) and $p$-value for each correlation measure at the bottom of the result tables. We set the significance level at $\alpha = 0.05$, and the significance for each measure in each replicate is determined after a Bonferroni correction ($< 5 \cdot 10^{-4}$). The following Case 1 shows that $G_2$ potentially regulates $G_1$, so $G_1$ expresses behind $G_2$.

**Case 1.**

$$G_{1i} = 2a \sin\left(\frac{T_i \pi}{42}\right) - 0.5 + c \times \varepsilon_{1i}, \tag{1}$$

$$G_{2i} = 2a \sin\left(\frac{(T_i - 7)\pi}{42}\right) - 0.5 + c \times \varepsilon_{2i}, \tag{2}$$

The true time-course expression of $G_1$ and $G_2$ (when $c = 0.0$) and a set of simulated data (when $c = 0.5, 1.0$, and 2.0) are plotted in Fig 2, in which the two dotted sine curves have a time difference of $\pi/6$.

As shown in Table 1, when $a = 0$ and noise in data increases from low, medium to high ($c = 0.5, 1.0$ to 2.0), the FPR of $K_c$-poly2 and $K_c$-poly3 are high (24-28% and 30-52%, respectively),

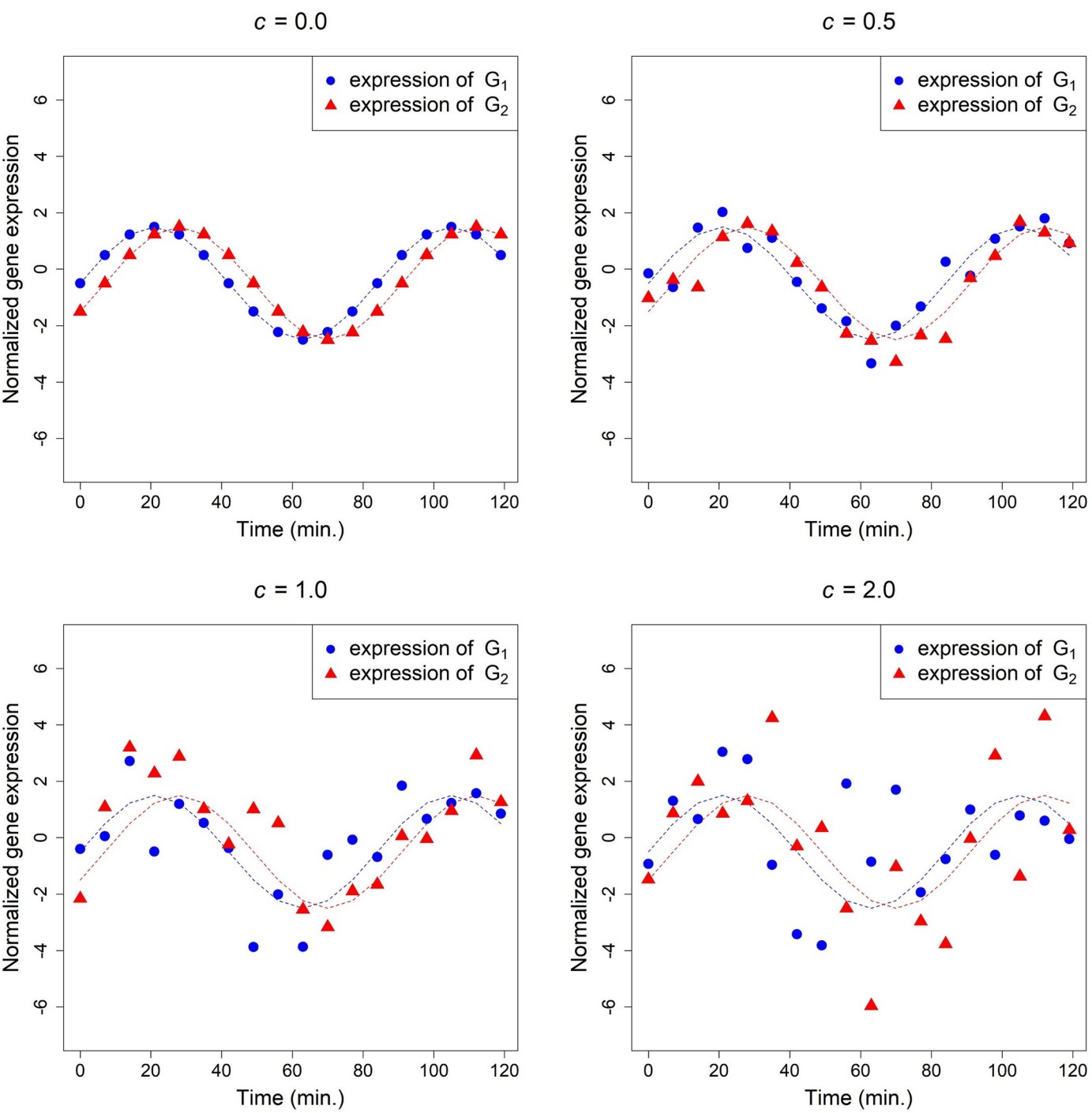

**Fig 2. The simulated expression of gene $G_1$ and $G_2$ generated by Eqs (1) and (2) with various values of $c$, where (▲) denotes the values of $G_{1i}$ ($G_{2i}$) and $i = 0, 1, \dots, 17$.**

while the FPR of Pearson's, Kendall's and distance correlation remain 0.0. When $a = 1$ and noise is low ($c = 0.5$), $K_c$-RBF, Pearson's correlation, and dCor have high TPRs (99%, 92%, and 92%, respectively), while Kendall's correlation has only 4%; $K_c$-RBF has a moderate FPR (18%). The advantage of $K_c$-RBF is clearer when noise is increased to moderate ($c = 1.0$), its

Table 1. Comparison of the six correlation measures on the nonlinear correlation between genes $G_1$ and $G_2$, generated by Eqs (1) and (2) with 100 replicates.

| $c =$ | 0.0 | 0.5 | 1.0 | 2.0 |
|---|---|---|---|---|
| **a = 0** | False positive rates | | | |
| $K_c$-poly2[a] | -- | 0.28 | 0.24 | 0.27 |
| $K_c$-poly3 | -- | 0.30 | 0.45 | 0.52 |
| $K_c$-RBF ($\gamma = 0.5$) | -- | 0.18 | 0.01 | 0.00 |
| Pearson's corr.[b] | -- | 0.00 | 0.00 | 0.00 |
| Kendall's corr. | -- | 0.00 | 0.00 | 0.00 |
| Distance corr. | -- | 0.00 | 0.00 | 0.00 |
| **a = 1** | True Positive rates | | | |
| $K_c$-poly2 | -- | 1.00 | 0.88 | 0.47 |
| $K_c$-poly3 | -- | 1.00 | 0.96 | 0.69 |
| $K_c$-RBF ($\gamma = 0.5$) | -- | 0.99 | 0.63 | 0.08 |
| Pearson's corr. | -- | 0.92 | 0.21 | 0.02 |
| Kendall's corr. | -- | 0.04 | 0.00 | 0.00 |
| Distance corr. | -- | 0.92 | 0.33 | 0.04 |
| | Correlation measurement (*p*-values) | | | |
| $K_c$-poly2 | 1.00 ($< \epsilon$[a]) | 0.99 ($< \epsilon$) | 0.88 ($< \epsilon$) | 0.47 (0.02) |
| $K_c$-poly3 | 1.00 ($< \epsilon$) | 1.00 ($< \epsilon$) | 0.96 ($< \epsilon$) | 0.67 (0.001) |
| $K_c$-RBF ($\gamma = 0.5$) | 0.97 ($< \epsilon$) | 0.92 ($< \epsilon$) | 0.71 ($< \epsilon$) | 0.32 (0.10) |
| Pearson's corr. | 0.86 ($< \epsilon$) | 0.78 ($< \epsilon$) | 0.61 (0.004) | 0.32 (0.10) |
| Kendall's corr. | 0.65 (0.002) | 0.57 (0.007) | 0.43 (0.04) | 0.20 (0.21) |
| Distance corr. | 0.84 ($< \epsilon$) | 0.79 ($< \epsilon$) | 0.65 (0.002) | 0.47 (0.02) |

[a]$K_c$-poly2, $K_c$-poly3, and $K_c$-RBF denote $K_c$ with the polynomial kernel of degree 2 and 3, and the RBF kernel, respectively.

[b]The notation corr. and $\epsilon$ denote correlation coefficient and $5 \cdot 10^{-4}$, respectively.

TPR is much higher (63%) than that of Pearson's correlation (21%), Kendall's correlation (0%) and dCor (33%), while its FPR (1%) is equivalent to that of Pearson's and Kendall's correlation and dCor (0%). Further, Fig 2 shows that when $c = 1.0$, $G_1$ and $G_2$ are nonlinearly correlated, $K_c$-RBF (0.71) is significant, but dCor, Pearson's and Kendall's correlation are not (their p-values are not $< 5 \cdot 10^{-4}$; Table 1).

The results show that $K_c$-RBF outperforms Pearson's and Kendall's correlation on the estimation of the nonlinear correlation in data generated with low to medium noise in Case 1. When noise is low, Pearson's correlation and dCor performs slightly better than $K_c$-RBF, $K_c$-RBF outperforms Pearson's correlation and dCor when noise is moderate. However, $K_c$ with polynomial kernels have high FPRs which may result in overestimation, and Kendall's correlation coefficients are small and insignificant, thus these measures are not suitable for estimation of nonlinear correlation.

**Case 2.**

$$G_{1i} = 2a \cos\left(\frac{(T_i - 21)\pi}{42}\right) + c \times \varepsilon_{1i}, \tag{3}$$

$$G_{2i} = a \cos\left(\frac{(T_i - 21)\pi}{42}\right) + c \times \varepsilon_{2i}. \tag{4}$$

The true time course expression of $G_1$ and $G_2$ (generated by Eqs (3) and (4) with $c = 0.0$), and a set of simulated data (generated by Eqs (3) and (4) with $c = 0.5, 1.0$ and $2.0$) are plotted in Fig 3, in which the dotted curves of the expression generated by $c = 0.0$ are two identical cosine curves except that the range of $G_1$ is twice that of $G_2$.

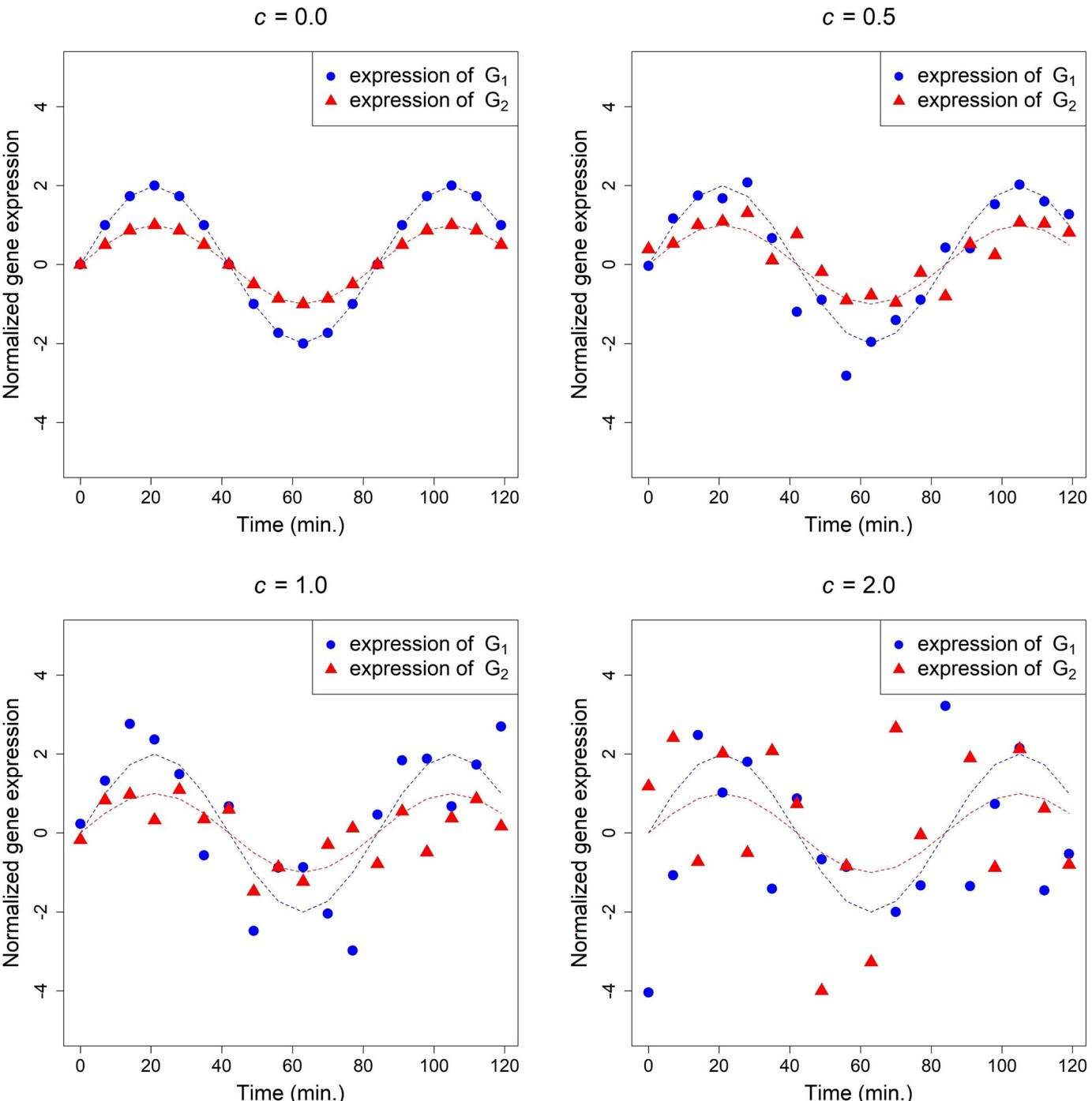

**Fig 3. The simulated expression of gene $G_1$ and $G_2$ generated by Eqs (3) and (4) with various values of $c$, where (•) (▲) denotes the values of $G_{1i}$ ($G_{2i}$) and $i = 0, 1, \ldots, 17$.**

The results are similar to those of Case 1 and are summarized in Table 2. Taking both PR and FPR into account, when noise is low, $K_c$-RBF performs similarly to Pearson's correlation and dCor on the estimation of the nonlinear correlation in Case 2. But $K_c$-RBF has better performance than Pearson's correlation and dCor when noise is moderate. Similar to Case 1, $K_c$ with polynomial kernels and Kendall's correlation coefficient are not suitable estimators, due to high FPRs and small and insignificant coefficients, respectively.

**Case 3.**

$$G_{1i} = 2a \cos(\frac{(T_i - 21)\pi}{42}) + c \times \varepsilon_{1i}, \tag{5}$$

$$G_{2i} = 2a \cos(\frac{(T_i - 21)\pi}{42}) + 3 + c \times \varepsilon_{2i}. \tag{6}$$

The true time course expression of gene $G_1$ and $G_2$ (generated by Eqs (5) and (6) with $c = 0$) and a set of simulated data (generated by Eqs (5) and (6) with $c = 0.5$, 1.0 and 2.0, respectively) are plotted in Fig 4. When $c = 0$, these two dotted expression curves differed only by a location shift of 3 in the y-axis, and as $c$ assumed larger values, two less identical curves were generated. The results are similar to those of Case 1 and 2 and are summarized in Table 3.

When the noise in data is low, Pearson's correlation and dCor perform slightly better than $K_c$-RBF (with smaller FPRs). However, when noise is moderate, $K_c$-RBF outperforms Pearson's

**Table 2. Comparison of the six correlation measures on the nonlinear correlation between $G_1$ and $G_2$, generated by Eqs (3) and (4) with 100 replicates.**

| $c =$ | 0.0 | 0.5 | 1.0 | 2.0 |
|---|---|---|---|---|
| **a = 0** | | False positive rates | | |
| $K_c$-poly2[a] | -- | 0.30 | 0.29 | 0.34 |
| $K_c$-poly3 | -- | 0.34 | 0.49 | 0.57 |
| $K_c$-RBF ($\gamma = 0.5$) | -- | 0.14 | 0.06 | 0.01 |
| Pearson's corr.[b] | -- | 0.00 | 0.00 | 0.00 |
| Kendall's corr. | -- | 0.00 | 0.00 | 0.00 |
| Distance corr. | -- | 0.00 | 0.00 | 0.00 |
| **a = 1** | | True Positive rates | | |
| $K_c$-poly2 | -- | 1.00 | 0.77 | 0.36 |
| $K_c$-poly3 | -- | 1.00 | 0.95 | 0.61 |
| $K_c$-RBF ($\gamma = 0.5$) | -- | 0.98 | 0.30 | 0.00 |
| Pearson's corr. | -- | 0.82 | 0.08 | 0.00 |
| Kendall's corr. | -- | 0.07 | 0.00 | 0.00 |
| Distance corr. | -- | 0.85 | 0.08 | 0.00 |
| | | Correlation measurement (p-values) | | |
| $K_c$-poly2 | 1.00 ($< \epsilon$ [a]) | 0.99 ($< \epsilon$) | 0.81 ($< \epsilon$) | 0.38 (0.06) |
| $K_c$-poly3 | 1.00 ($< \epsilon$) | 1.00 ($< \epsilon$) | 0.88 ($< \epsilon$) | 0.49 (0.02) |
| $K_c$-RBF ($\gamma = 0.5$) | 1.00 ($< \epsilon$) | 0.92 ($< \epsilon$) | 0.55 (0.01) | 0.18 (0.24) |
| Pearson's corr. | 1.00 ($< \epsilon$) | 0.77 ($< \epsilon$) | 0.47 (0.03) | 0.20 (0.21) |
| Kendall's corr. | 1.00 ($< \epsilon$) | 0.57 (0.01) | 0.33 (0.09) | 0.13 (0.30) |
| Distance corr. | 1.00 ($< \epsilon$) | 0.79 ($< \epsilon$) | 0.55 (0.01) | 0.41 (0.05) |

[a]$K_c$-poly2, $K_c$-poly3, and $K_c$-RBF denote $K_c$ with the polynomial kernel of degree 2 and 3, and the RBF kernel, respectively.

[b]The notation corr. and $\epsilon$ denote correlation coefficient and $5 \cdot 10^{-4}$, respectively.

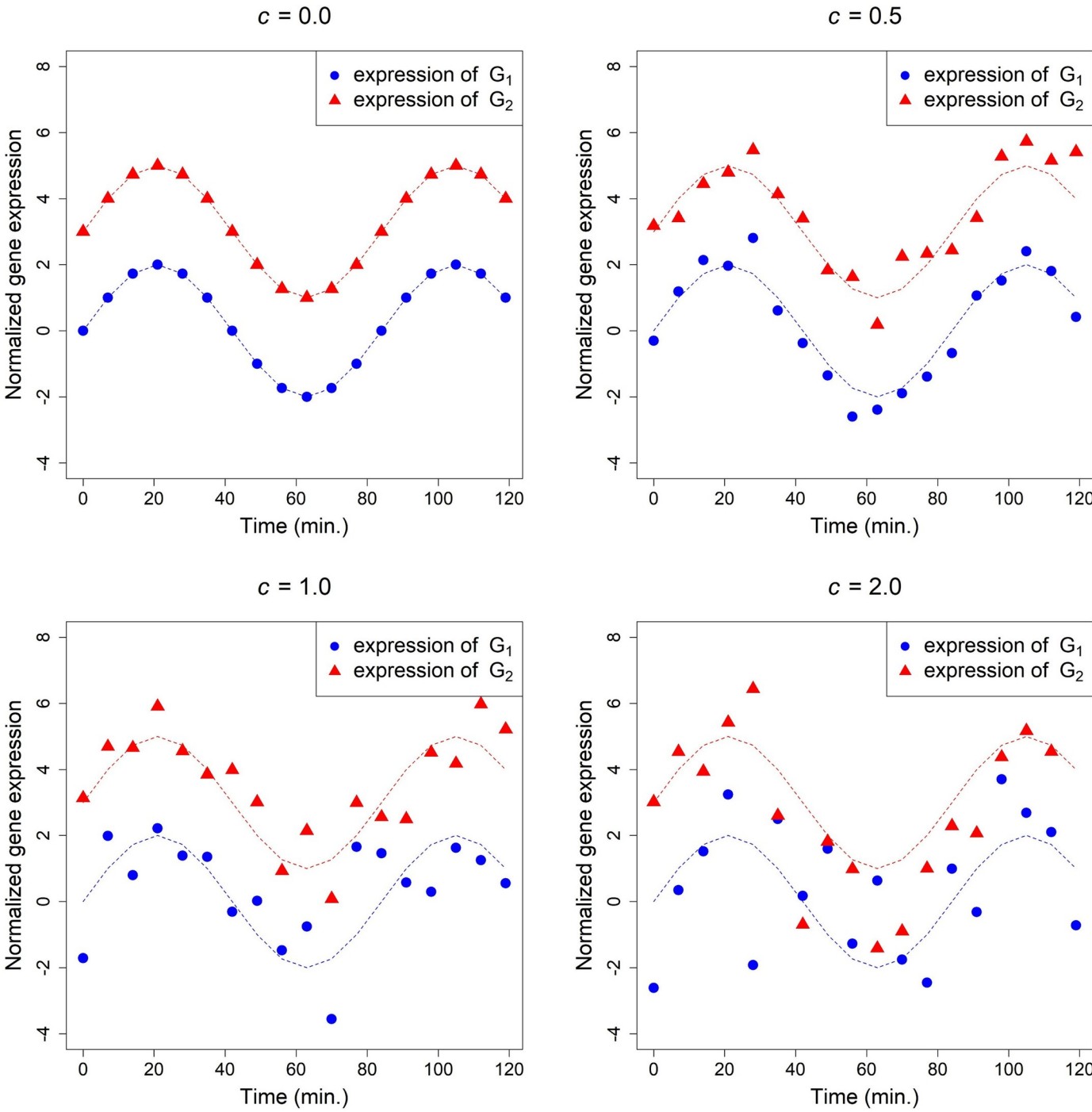

**Fig 4. The simulated expression of gene $G_1$ and $G_2$ generated by Eqs (5) and (6) with various values of $c$, where (▲) denotes the values of $G_{1i}$ ($G_{2i}$) and $i = 0, 1, \ldots, 17$.**

correlation and dCor on the estimation of the nonlinear correlation in Case 3. And similar to Case 1 and 2, $K_c$ with polynomial kernels and Kendall's correlation are not suitable estimators, due to high FPRs and small and insignificant coefficients, respectively, when noise is low to moderate in data.

**Table 3. Comparison of the six correlation measures on the nonlinear correlation between genes $G_1$ and $G_2$, generated by Eqs (5) and (6) with 100 replicates.**

| $c =$ | 0.0 | 0.5 | 1.0 | 2.0 |
|---|---|---|---|---|
| **a = 0** | | False positive rates | | |
| $K_c$-poly2[a] | -- | 0.20 | 0.19 | 0.23 |
| $K_c$-poly3 | -- | 0.26 | 0.40 | 0.49 |
| $K_c$-RBF ($\gamma = 0.5$) | -- | 0.11 | 0.07 | 0.01 |
| Pearson's corr.[b] | -- | 0.00 | 0.00 | 0.00 |
| Kendall's corr. | -- | 0.00 | 0.00 | 0.00 |
| Distance corr. | -- | 0.00 | 0.00 | 0.00 |
| **a = 1** | | True Positive rates | | |
| $K_c$-poly2 | -- | 1.00 | 0.94 | 0.42 |
| $K_c$-poly3 | -- | 1.00 | 0.99 | 0.75 |
| $K_c$-RBF ($\gamma = 0.5$) | -- | 1.00 | 0.76 | 0.10 |
| Pearson's corr. | -- | 1.00 | 0.39 | 0.02 |
| Kendall's corr. | -- | 0.49 | 0.02 | 0.00 |
| Distance corr. | -- | 1.00 | 0.54 | 0.05 |
| | | Correlation measurement (p-values) | | |
| $K_c$-poly2 | 1.00 ($< \epsilon$ [a]) | 1.00 ($< \epsilon$) | 0.92 ($< \epsilon$) | 0.48 (0.02) |
| $K_c$-poly3 | 1.00 ($< \epsilon$) | 1.00 ($< \epsilon$) | 0.98 ($< \epsilon$) | 0.72 ($< \epsilon$) |
| $K_c$-RBF ($\gamma = 0.5$) | 1.00 ($< \epsilon$) | 0.97 ($< \epsilon$) | 0.78 ($< \epsilon$) | 0.37 (0.07) |
| Pearson's corr. | 1.00 ($< \epsilon$) | 0.90 ($< \epsilon$) | 0.68 (0.001) | 0.35 (0.08) |
| Kendall's corr. | 1.00 ($< \epsilon$) | 0.70 (0.001) | 0.49 (0.02) | 0.23 (0.18) |
| Distance corr. | 1.00 ($< \epsilon$) | 0.90 ($< \epsilon$) | 0.71 ($< \epsilon$) | 0.47 (0.02) |

[a]$K_c$-poly2, $K_c$-poly3, and $K_c$-RBF denote $K_c$ with the polynomial kernel of degree 2 and 3, and the RBF kernel, respectively.

[b]The notation corr. and $\epsilon$ denote correlation coefficient and $5 \cdot 10^{-4}$, respectively.

## Applications

In this section, we first applied the proposed kernelization correlation procedure to identify genes whose expression patterns across time were similar or complementary to *IL17A*, which played an important role in the early differentiation of Th17 cells in humans. Through this procedure, the genes significantly correlated to *IL17A* were inferred to be involved in the early differentiation of Th17 cells. This is a simple but efficient way to identify genes associated with the differentiation of Th17 cells. Next, we applied $K_c$ to estimate the nonlinear correlation of cell cycle genes in yeast.

**Application 1.** The expression of *IL17A* is commonly used to assess the Th17 polarization efficiency [16]. Previously, transcriptional factors Rorc and Stat3 were revealed to be the key regulators at the early stage of Th17 differentiation in murine and humans [22]. Aijo and colleagues profiled gene expression in the early phase of Th17 differentiation to understand the process of differentiation and how the differentiation signal propagates through various pathways. The knowledge they gained is very useful for uncovering markers of differentiation of Th17 cell populations.

In this application, we applied $K_c$ to the RNA-seq data of the early phase of Th17 cell differentiation and T-cell activation (Th0) cells [8]. There are three replicates for each gene profiled from Th0 and Th17 cells at 0, 12, 24, 48 and 72 h, in which the differences in differentiation efficiency among the replicates have been adjusted; see Section 3.2 [8].

The time-course expression of *IL17A* and *RORC* (*ISG20*, *RAB3*, and *TIAM1*) in Fig 1 (Fig 7) of [8], respectively, show that *IL17A* has a positive correlation with *RORC*, *ISG20*, and *RAB3*, but it has a negative correlation with *TIAM1*. These correlations are consistent with the expression (in normalized read counts) of these genes in Th17 cells. Moreover, the CoS test was applied (with 1,000 repeats) to test for the nonlinearity of the four *IL17A* gene pairs, which were all significant with *P* equal to 0.014 and 0.000 (rounded to the third digit) for the latter three pairs, respectively. This result justifies that the nonlinear correlations of these pairs exist.

We applied $K_c$-RBF (with the default $\gamma$ value) to estimate the nonlinear correlation of similar-patterned (complementary-patterned) gene pairs, e.g., *IL17A-ISG20* (*IL17A-TIAM1*) profiled in Th17 cells [8]. Because the normalized data of these genes in replicate 1 differed much from those of replicate 2 and 3, we set replicate 1 data aside and computed $K_c$ using replicate 2 and 3 data. For genes whose expression of Th0 cells was highly similar to that of Th17 cells, they were not involved in the differentiation of Th17 cells. Thus, to exclude genes irrelevant to immune differentiation, we first computed $K_c$ of *MAP1B*, *RORC*, *KIF11*, *IGS20*, *RAB3* and *TIAM1* with themselves in Th17 cells and Th0 cells, e.g., correlation of the expression of *MAP1B* in Th17 and the expression of *MAP1B* in Th0 cells. Since self-correlation is similar-patterned, we used $K_c$-RBF ($\gamma = 0.5$), and obtained an averaged self-correlation of *MAP1B* (*KIF11*) equal to 0.998 (0.996) with $P < 0.001$, using normalized read counts (via DESeq). This result is consistent with the expression of *MAP1B* (*KIF11*) in Th0 cells being highly similar to that in Th17 cells (S4 Fig, [8]). Consequently, *MAP1B* and *KIF11* are likely to be false positives (not involved in immune differentiation) and should be excluded.

Next, we applied $K_c$-RBF to compute the correlation of *IL17A* with each of *TIAM1*, *ISG20*, *RAB3* and *RORC* in Th17 cells. For similar-patterned gene pairs, the default value of $\gamma$ (0.5) was used. For negatively correlated gene pairs such as *IL17A-TIAM1*, we first used replicate 2 (replicate 3) gene expression to train $\gamma$, then applied the trained $K_c$ to estimate the nonlinear correlation for replicate 3 (replicate 2) data and averaged the two values of $K_c$. The cross-validation formula to train $\gamma$ was $CV = \underset{\gamma}{\arg\min} \frac{1}{2} \sum_{i=1}^{2} (K_{ci}(\gamma) - \overline{K_c}(\gamma))^2$, where $\overline{K_c}(\gamma) = \frac{1}{2} \sum_{i=1}^{2} K_{ci}(\gamma)$, and the trained $\gamma = 7.5$ for *IL17A-TIAM1*. The averaged $K_c$ value of *IL17A-TIAM1*, *IL17A-ISG20*, *IL17A-RAB3* and *IL17A-RORC* are -0.21, 0.83, 0.90 and 0.84, respectively and the corresponding p-values using t-test are 0.002, 0.020, 0.006 and 0.017, which are all significant at $\alpha = 0.05$. These four genes have significant $K_c$ with *IL17A*, thus these are classified as involved in immune differentiation.

*Comparison of $K_c$ to DyNB, timepoint-wise analysis using DESeq and dCor.* Aijo and colleagues showed that DyNB [8] was able to uncover the nonlinear correlation of *IL17A* with *ISG20*, *RAB3* and *TIAM1*, which DESeq [20] from timepoint-wise analysis failed to uncover; mRNA expression of *IL17A* and *RORC* also exhibited similar patterns across time (Fig 1(B) and 1(D) in [8]). Further, timepoint-wise analysis (DESeq) detected two false positives, *KIF11* and *MAP1B*. We note that timepoint-wise analysis did not take into account correlations between time points and the whole pattern of time-course gene expression. Therefore, both $K_c$ and DyNB could detect the aforementioned nonlinear temporal correlation of the above four pairs, but not DESeq. We further compared the nonlinear correlation measure dCor to $K_c$. DCor estimated the correlation of *IL17A-IGS20* and *IL17A-RORC* correctly (at the level of 0.05), but failed to quantify *IL17A-RAB3* and *IL17A-TIAM1*. As *IL17A-TIAM1* exhibited a complementary pattern (Fig 7(C) in [8]), their correlation should be negative; we refer to Table 4 for details.

**Application 2.** In this application, we applied Pearson's correlation, $K_c$ and dCor [11] to estimate the nonlinear correlations of six gene pairs, which were involved in the cell cycle of *S*.

**Table 4. DCor and $K_c$-RBF estimated for the four gene pairs of *IL17A*, which played an important role in the early differentiation of Th17 cells in humans.**

| mean (se) p-value | dCor | $K_c$-RBF[a] |
|---|---|---|
| *IL17A-TIAM1* | 0.65 (0.09) 0.032 | -0.21 (0.001) 0.002 |
| *IL17A-ISG20* | 0.61 (0.10) 0.035 | 0.83 (0.04) 0.020 |
| *IL17A-RAB3* | 0.66 (0.24) 0.081 | 0.90 (0.01) 0.006 |
| *IL17A-RORC* | 0.68 (0.15) 0.048 | 0.84 (0.03) 0.017 |

[a]The trained $\gamma = 7.5$ ($\gamma = 0.5$) was used in the RBF kernel for *IL17A-TIAM1* (the remaining three pairs).

*cerevisiae* [9]. These pairs are *RNR1-SWE1*, *RNR1-RAD51*, *SWE1-RAD51*, *HST3-RAD51*, *HST3-RNR1*, and *HST3-SWE1*, where the former three pairs have similar patterns, but the latter three exhibit complementary patterns. We first tested whether the nonlinear correlations exist in these pairs by the CoS test [12]. The *P* value of CoS (with 1,000 repeats) for these pairs was equal to 0.000, 0.000, 0.004, 0.000, 0.002, and 0.003, respectively, which demonstrated nonlinearity existed in these pairs. Next, we applied $K_c$-RBF (with the default $\gamma = 0.5$) to estimate the nonlinear correlation of gene pairs with similar patterns, similar to Application 1. We treated data of the two cell cycles in the experiments, namely time $t$ = 1-9 and 10-18 (two biological repeats), as two replicates. The Pearson's correlation, dCor and $K_c$ computed by the replicates for (*RNR1-SWE1*, *RNR1-RAD51* and *SWE1-RAD51)* were (0.82, 0.92 and 0.72), (0.85, 0.91 and 0.78) and (0.91, 0.99 and 0.91), respectively. All of these were significant at the 0.10 level; see Table 5 for details. For gene pairs with anti-similar patterns, we used replicate 1 (replicate 2) to train $\gamma$ of the RBF kernel and applied the trained $K_c$ to estimate the nonlinear correlation for replicate 2 (replicate 1). The objective function to be optimized is as follows.

$$CV = \arg\min_{\gamma} \frac{1}{2}\sum_{i=1}^{2}(K_{ci}(\gamma) - \overline{K_c}(\gamma))^2, \text{ where } \overline{K_c}(\gamma) = \frac{1}{2}\sum_{i=1}^{2}K_{ci}(\gamma),$$

and $K_{ci}$ is the value of $K_c$ in biological repeat $i$.

As shown in the S1 Dataset in which $\gamma$ was trained, the values of the objective function with $\gamma$ =1.0, 1.0, and 0.7 did not differ much from the associated global minimum of *HST3-RAD51*, *RNR1-HST3*, and *HST3-SWE1*, respectively; the differences were $< 8\times10^{-5}$, thus we used $K_c$-RBF with these trained $\gamma$ values. The estimated correlation of *RNR1-HST3*, *HST3-RAD51* and *HST3-SWE1* by Pearson's correlation ($r$), dCor and $K_c$ are summarized in the lower half of

**Table 5. Pearson's $r$, dCor, and $K_c$-RBF estimated for the pairs of similar and complementary patterned cell cycle genes in yeast.**

| mean (se) p-value | Pearson's $r$ | dCor | $K_c$-RBF[a] |
|---|---|---|---|
| Similar patterned | | | |
| *RNR1-SWE1* | 0.82 (0.06) 0.048 | 0.85 (0.01) 0.003 | 0.91 (0.07)0.050 |
| *RNR1-RAD51* | 0.92 (0.04)0.024 | 0.91 (0.06) 0.014 | 0.99 (0.004) 0.003 |
| *SWE1-RAD51* | 0.72 (0.11) 0.094 | 0.78 (0.15) 0.043 | 0.91 (0.04) 0.031 |
| Complementary patterned | | | |
| *HST3-RNR1* | -0.55 (0.16) 0.129 | 0.72 (0.01) 0.003 | -0.85 (0.01) 0.006 |
| *HST3-RAD51* | -0.50 (0.23) 0.203 | 0.65 (0.06) 0.022 | -0.87 (0.004) 0.002 |
| *HST3-SWE1* | -0.49 (0.26) 0.224 | 0.67 (0.01) 0.003 | -0.75 (0.11) 0.066 |

[a]The default $\gamma = 0.5$ was used in the RBF kernel for the similar patterned pairs. The trained $\gamma = 1.0$ ($\gamma = 0.7$) was used in the RBF kernel for *HST3-RNR1* and *HST3-RAD51* (*HST3-SWE1*).

Table 5. Note that this training of the $\gamma$ value is essential for complementary patterned variables.

At the 0.10 significance level, $K_c$ of *HST3-RNR1*, *HST3-RAD51*, and *HST3-SWE1* were all significant, but none of the Pearson's correlation coefficients was significant, nor did dCor result in negative signs of the complementary patterns correctly. In particular, both Pearson's correlation and dCor did not reflect the complementary pattern of *HST3-RAD51* as shown in Fig 1A.

## A rule of thumb for kernel selection of $K_c$

The simulation study shows that when the noise of data is low to moderate, taking both TPR and FPR into account (e.g., using the likelihood ratio for a positive test TPR/FPR), $K_c$-RBF is better than $K_c$-poly2, and $K_c$-poly3 performs the worst. Furthermore, Application 1 and 2 show that $K_c$ -RBF with a trained $\gamma$ value ($K_c$-RBF with $\gamma = 0.5$) is able to detect negative (positive) correlations adequately, but $K_c$-poly2 fails to detect *IL17A-TIAM1* (-0.0002 with $P = 0.500$) and *IL17A-RORC* (-0.0002 with $P = 0.500$). Therefore, we suggest using $K_c$-RBF (with $\gamma = 0.5$) for positively correlated variables, and $K_c$ -RBF (with a trained $\gamma$ value) for negatively correlated variables.

## Discussion

We have proposed the kernelized correlation $K_c$, which has been shown able to detect a nonlinear correlation of two variables (across time) as shown in the simulation study and two applications. To the best of our knowledge, this nonlinear measure is the first to quantify negatively correlated pairwise variables by a negative value, and it can also be applied to data exploration, variable selection, and others. Nevertheless, it cannot be applied to measure correlations of paired variables in static experiments, e.g., gene A in the control and experimental groups. The advantages of $K_c$ lie in that it is simple and distribution-free. This tool will enable non-computational researchers to identify nonlinear correlations in biomedical data which may have important applications, such as identifying genes involved in the differentiation of human T helper cells.

For similar patterned pairwise variables, using the default $\gamma$ value of the RBF kernel is sufficient. For complementary patterned pairwise variables, a CV-trained value of $\gamma$ is required similar to other bandwidth-based methods; nevertheless, this training step is fairly simple using the cross-validation formula, and the code is also provided at http://staff.stat.sinica.edu.tw/gshieh/KC/Kc.html.

As demonstrated in the Result section, the proposed $K_c$ was able to quantify nonlinear associations in the simulated cases that had been designed with some degree of nonlinear correlation, in contrast to Pearson's and Kendall's correlations, which did not detect significant correlation when the data were nonlinear. In Application 1, $K_c$-RBF was shown to identify the negative correlation of *IL17A-TIAM1* and the remaining three gene pairs with positive correlations, as well as the sophisticated DyNB [3], while DESeq failed in these pairs and dCor only identified *IL17A-ISG20* and *IL17A-RORC* correctly. In Application 2, $K_c$ was shown to detect negative nonlinear correlations of the three gene pairs involved in yeast cell cycle regulation, while both Pearson's correlation and dCor failed in all these pairs.

Our method is a general approach to detecting nonlinear correlations. The property of distribution-free makes $K_c$ applicable to a wide variety of problems. Although we only applied it to gene expression data in this study, this method can be applied to other types of omic data, e.g., proteomics and metabolomics. Taken together, this nonlinear correlation approach is useful for the estimation of the nonlinearity of biological associations. In future research, a natural

extension would be to develop a multivariate version of $K_c$. The derivation of measures for nonlinear correlation in functional data is another promising topic.

## Supporting information

**S1 File. Training the parameter gamma in the RBF kernel of Application 1 and 2.** (DOCX)

**S2 File. The R code of kernelized correlation $K_c$.** (RMD)

**S1 Dataset. Dataset of Application 1.** (ZIP)

**S2 Dataset. Dataset of Application 2.** (CSV)

## Acknowledgments

We are grateful to the academic editor and two anonymous reviewers for the constructive comments. We thank Chen-Ming Chiou and Su-Yun Huang for their helpful discussions. We are indebted to Jia-Hua Tsai, Yu-Ping Tseng, and Ce-Bo Yang for the computational work.

## Author Contributions

**Conceptualization:** Grace S. Shieh.

**Formal analysis:** Suneel Babu Chatla.

**Funding acquisition:** Grace S. Shieh.

**Investigation:** Li-Shan Huang, Grace S. Shieh.

**Methodology:** Yogesh M. Tripathi, Yuan-Chin I. Chang.

**Software:** Li-Shan Huang.

**Supervision:** Li-Shan Huang, Grace S. Shieh.

**Validation:** Grace S. Shieh.

**Writing – original draft:** Yogesh M. Tripathi, Grace S. Shieh.

**Writing – review & editing:** Li-Shan Huang, Grace S. Shieh.

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
