## [Decision Letter · Decision Letter 0]

17 May 2022

PONE-D-21-31352A Nonlinear Correlation Measure with Applications to Gene Expression DataPLOS ONE

Dear Dr. Shieh,

Thank you for submitting your manuscript to PLOS ONE. After careful consideration, we feel that it has merit but does not fully meet PLOS ONE’s publication criteria as it currently stands. Therefore, we invite you to submit a revised version of the manuscript that addresses the points raised during the review process.

We look forward to receiving your revised manuscript.

Kind regards,

Maria Alessandra Ragusa, PhD Professor

Academic Editor

PLOS ONE

**Journal requirements:**

https://journals.plos.org/plosone/s/file?id=ba62/PLOSOne_formatting_sample_title_authors_affiliations.pdf".

**Additional Editor Comments:**

Dear Corresponding Author,

according to the referee's comments the paper is accepted for publication.

Best regards.

**Reviewers' comments:**

Reviewer's Responses to Questions

**Comments to the Author**

1. Is the manuscript technically sound, and do the data support the conclusions?

Reviewer #1: Yes

Reviewer #2: Yes

2. Has the statistical analysis been performed appropriately and rigorously? 

Reviewer #1: Yes

Reviewer #2: Yes

3. Have the authors made all data underlying the findings in their manuscript fully available?

Reviewer #1: Yes

Reviewer #2: Yes

4. Is the manuscript presented in an intelligible fashion and written in standard English?

Reviewer #1: Yes

Reviewer #2: Yes

5. Review Comments to the Author

Reviewer #1: The manuscript presents a novel approach to reveal nonlinear correlation in biomedical datasets.

The nonlinear kernel measure has been proposed to project high-dimensional data into Hilbert space. The authors demonstrated the effectiveness their approach for discovery of genes involved in differentiation human T helper cells. The statistical analysis provides sufficient evidence for advantages of the approach with respect to several canonical correlation measures.

Reviewer #2: The authors proposed a kernelized correlation coefficient to measure a possibly nonlinear association among genes. Although the proposed method is straightforward, it has potential and broad applicability in bioinformatics. My specific comments are given in the following.

- One obvious competitor is distance correlation to measure the nonlinear association between two variables. So I suggest adding this method to the comparison.

- I believe the choice of the kernel is quite essential, especially in the applications in genetics. I suggest providing more details about how to choose the kernel in practice.

6. PLOS authors have the option to publish the peer review history of their article (what does this mean?). If published, this will include your full peer review and any attached files.

Reviewer #1: No

Reviewer #2: No

---

## [Author Response · Author response to Decision Letter 0]

1 Jun 2022

We have included point-to-point replies to the comments and suggestions made by AE and Reviewers

in the "Response to Reviewers" .docx

---

## [Editor Report · Decision Letter 1]

8 Jun 2022

A Nonlinear Correlation Measure with Applications to Gene Expression Data

PONE-D-21-31352R1

Dear Dr. Shieh,

We’re pleased to inform you that your manuscript has been judged scientifically suitable for publication and will be formally accepted for publication once it meets all outstanding technical requirements.

Kind regards,

Maria Alessandra Ragusa, PhD Professor

Academic Editor

PLOS ONE

Additional Editor Comments (optional):

The revised version is now ready for publication.

Best regards.
---

## [Editor Report · Acceptance letter]

10 Jun 2022

PONE-D-21-31352R1 

A nonlinear correlation measure with applications to gene expression data 

Dear Dr. Shieh:

I'm pleased to inform you that your manuscript has been deemed suitable for publication in PLOS ONE. Congratulations! Your manuscript is now with our production department. 

Kind regards, 

on behalf of

Dr. Maria Alessandra Ragusa 

Academic Editor

PLOS ONE